# Numerical Simulation and Experimental Analysis of Dynamic Continuous Operation of Low-Concentration Coalbed-Methane-Mixing Device

Lu Xiao [1,2,3]

1   Research Center of Fluid Machinery Engineering and Technology, Jiangsu University,
    Zhenjiang 212013, China; xiaolu8317@126.com
2   China Coal Technology Engineering Group Chongqing Research Institute, Chongqing 400037, China
3   National Key Laboratory of Gas Disaster Detecting Preventing and Emergency Controlling,
    Chongqing 400037, China

**Abstract:** The concentration of low-concentration coalbed methane extracted from underground coal mine fluctuates greatly, which does not meet the requirements of intake concentration of coalbed-methane utilization devices. Due to this fluctuation, the coalbed-methane-utilization device cannot maintain stable and safe operation. The gas-mixing device is mainly used in coalbed-methane-utilization systems, providing each with a stable feed gas source with qualified concentration. In order to solve the problems of unsatisfactory uniformity of gas mixing and the large resistance of the existing coalbed-methane-mixing device, the mathematical model of the internal flow of the gas-mixing device is established. The influence of the internal structure of the gas-mixing device on the change in the uniformity of gas mixing and resistance loss is studied by numerical simulation and experiment. When the flow is 7000 Nm$^3$/h, 50,000 Nm$^3$/h and 160,000 Nm$^3$/h, respectively, the spiral structure combination of L-N-R, N-L-R and L-N-R is optimal (L, R, N indicate left rotation, right rotation and without setting screw, respectively). There are some defects in the processing technology of the experimental device, which make the simulation and experimental data different. The numerical simulation of the gas-mixing process inside the unit can provide technical means for reducing resistance and improving the uniformity of gas mixing, and provide a stable gas source and safety measures for the coalbed-methane-utilization unit.

**Keywords:** low concentration coalbed methane; mixing device; numerical simulation; uniformity; resistance loss

## 1. Introduction

The main component of coalbed methane is methane, which is a strong greenhouse gas [1]. Its greenhouse gas effect is 25 times that of carbon dioxide [2]. At the same time, coalbed methane is an important unconventional natural gas resource [3]. Extracting and utilizing coalbed methane not only can reduce coal mine disasters, but also reduce greenhouse gas emissions [4]. The world's coalbed methane reserves are about 240 trillion cubic meters, and the huge reserves are an effective supplement to conventional natural gas resources [5]. Coalbed methane is an excellent source of clean energy, and its low utilization rate is a common problem [6]. In the utilization of coalbed methane [7], low-concentration coalbed methane with methane concentration of 10% to 30% is mainly used for power generation, while ultra-low-concentration coalbed methane with a concentration of less than 10% is mixed with the wind-draining gas (or air), so that the concentration of the mixed gas is stabilized after 1.2%, and then enters into the thermal storage oxidation device to provide heat for the coal mine [8]. The direct consequence of a low coalbed methane utilization rate is large methane emissions and serious environmental pollution [9].

Coal enterprises need to use a lot of energy, but also produce a lot of $CO_2$. They must vigorously develop new energy resources and take the road of green and low-carbon development and utilization innovation [10,11]. The utilization of coalbed methane can significantly reduce carbon emissions compared with direct release [12]. The application prospects of coalbed methane cascade utilization technology are bright.

Coalbed methane utilization is faced with the main issues of an unstable gas source concentration and its inflammable and explosive nature [13]. Among them, the large fluctuation of methane concentration of coalbed methane is an important factor restricting the utilization of coalbed methane. The Chinese standard (*AQ1075-2009*) requires the low-concentration coalbed-methane power-generation device change the range of methane concentration in feed gas within 30 s, and that it shall not exceed 2%. If it exceeds this range, it will be shut down. However, the actual extracted coalbed methane concentration fluctuates greatly, which directly reduces the efficiency of the generator. In the thermal storage oxidation unit, too low a concentration will cause insufficient heat generated by the unit and the inability to maintain the heat balance of its own operation, and will eventually lead to furnace shutdown [14] while too high concentration may cause explosions [15].

In general, the concentration of coalbed methane extracted from underground coal mines is does not match with the intake concentration required by the backend coalbed-methane-utilization device. Coalbed-methane-mixing devices are mainly used in coalbed-methane heat-storage oxidation devices and coalbed-methane power-generation devices, providing each with a stable feed gas source with a concentration that meets the requirements [16]. The two most important parameters for evaluating the performance of gas-mixing devices are the resistance and uniformity of the mixed gas [17]. Coalbed methane is mainly pumped from a fractured coal seam by a water ring vacuum pump [18], and the pressure is relatively low, usually below 10 kPa. However, due to the long transmission pipeline and the large resistance of the transmission pipeline, the pressure at the front end of the gas-mixing device is only about 5000 Pa [19]. If the resistance of the gas-mixing device cannot be reduced, the continuous and stable operation of the coalbed-methane-utilization device at the backend cannot be guaranteed. The thermal storage oxidation unit requires the average concentration of methane in the feed gas at the cross-section to be maintained at about 1.2% [20]. Due to the low concentration and pressure, the cross-sectional area of the intake pipeline is large, and most of the diameters are more than 1 m. If the uniformity of the gas-mixing device is not enough, it is very likely that the methane concentration in some areas of the cross-section of the pipeline will be greater than 5% (the lower limit of possible methane explosion), resulting in the explosion of the heat-storage oxidation device. According to experience, it is generally required that the resistance of a coalbed-methane-mixing device shall not be greater than 500 Pa and the uniformity shall not be less than 90%.

At present, coal mines outside China make less use of coalbed methane. Most coal mines in China generally do not set up special gas-mixing devices when using coalbed methane, but use a large container instead. No structure is set inside the container to disturb the flow of gas. The two streams of coalbed methane are only mixed naturally in the container, resulting in poor uniformity of gas concentration at the outlet section. In the early stage, the author set up turbulence blades in the container according to experience, and the uniformity was improved. However, the resistance of the processed air mixing device was generally more than 1500 Pa in the actual operation. Resistance and uniformity are a pair of contradictory performance parameters. This paper studies the construction method of the three-dimensional calculation model of the internal flow and gas-mixing process of the low-concentration coalbed-methane-mixing device, reveals the influence of the internal structure of the gas-mixing device on the gas-mixing uniformity and resistance loss, and discusses a new method of device optimization design through the simulation calculation and experimental verification of structural parameters. From this, the optimal scheme of spiral blade structure combination in the device is obtained.

## 2. Computational Model

### 2.1. Geometric Model Construction

In this paper, a given geometric model and experimental data are used in the geometric dimension calculation process of three sets of different flow device models. The calculation process adopts UG software for three-dimensional modeling, HyperMesh software for grid division, and uses a fluent module in ANSYS for relevant calculation and post-processing analysis. Figure 1 shows the geometric shape and corresponding dimensions of a 7000 Nm$^3$/h gas-mixing device. This scale of this gas-mixing device is widely used in low-concentration coalbed-methane power-generation devices in coal mines. For the convenience of explanation, the main areas inside the device are named, as shown in Figure 2. The outlet of the high-concentration methane pipeline to the outlet of the mixing device is defined as the mixed zone, the inside of the high-concentration methane pipeline is defined as the tubule spiral region and the inlet of the device flange to the outlet of the high-concentration methane pipeline is defined as the large tube spiral region.

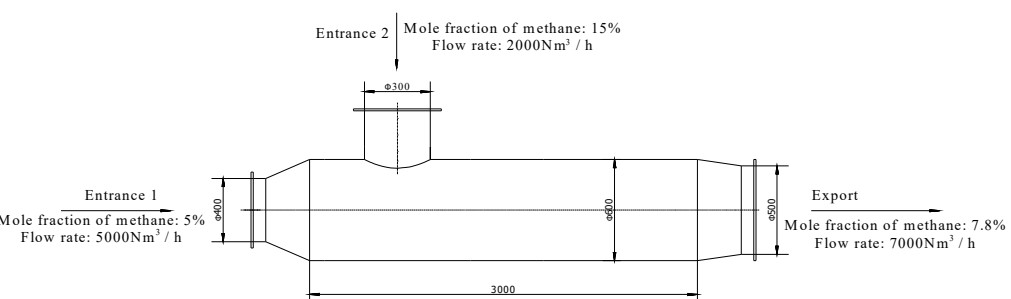

**Figure 1.** Model framework of 7000 Nm$^3$/h gas-mixing device.

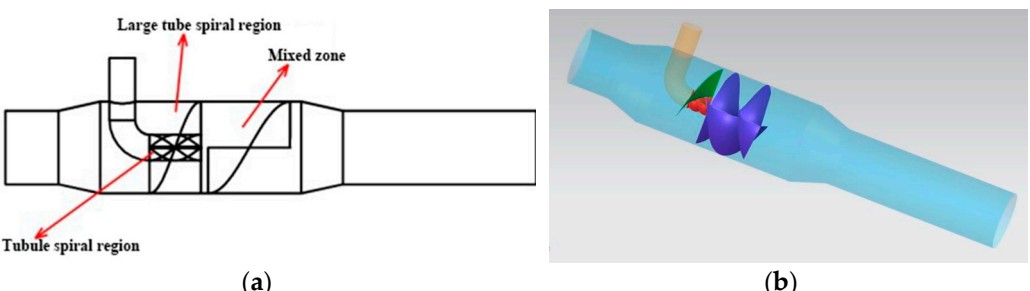

| (**a**) | (**b**) |

**Figure 2.** Main areas of the device, labeled. (**a**) Area diagram, (**b**) Blade diagram of 3 areas.

The gas in the large pipe area and the small pipe area is separated by the pipe wall of the small pipe area. Before entering the mixing area, the two gases do not contact. After entering the mixing area, they begin to contact and mix. The small pipe extends radially into the center of the large pipe area from the hole on the pipe wall of the large pipe, and a 90° elbow is used to change the direction from radial to axial, and spiral blades are set from the end of the elbow.

The geometric model of each component of the gas-mixing device is constructed by UG software. Firstly, the helix is directly inserted into the corresponding position of the device. After inputting the parameters such as helix diameter, pitch, number of helix turns and rotation direction through the command panel, the software will directly generate the corresponding curve. The calculation formula of helix total length (*L*) is: pitch (*P*) multiplied by number of turns (*n*). Following this, the spiral sheet of the helix is stretched through the "scan" command to produce the preliminary two-dimensional model of the helix. Finally, the required solid helix can be obtained by thickening the sheet. The spiral thickness used in the calculation is 6 mm.

In the simulation process, I found that there is obvious reflux at the outlet of the device, which has a great impact on the convergence of the calculation results. The main reason

is that the outlet position required for the calculation is just the cut-off position of the pipeline. In the simulation process, we found that there is obvious reflux at the outlet of the device, which has a great impact on the convergence of the calculation results. The main reason is that the outlet position required for the calculation is just the cut-off position of the pipeline. The vortex generated by gas mixing in the pipeline is not completely in the calculation domain at the outlet, which affects the free flow of the working medium in the fluid domain. The front end of the vortex outlet is a shrinking body, and the downstream is connected to a straight cylindrical pipe. The streamline of the fluid will change at this outlet. If the calculation is stopped suddenly, the convergence of FLUENT calculation may be affected, resulting in inaccurate calculation results. Thus, all calculation domain models in this paper set the outlet extension of 6 m and the inlet extension of 1 m. In addition, considering that the existence of the screw holes in the actual model will affect the mesh division of the model and the quality of the mesh, the geometric model adopted in this simulation is set as seamless connection between the spiral blades and the pipe wall, and the blades are connected by a solid shaft.

*2.2. Network Division*

First, we compare the tetrahedral meshing method with the tetrahedral hexahedral hybrid meshing method, and find that the stable component fluctuation value of the pure tetrahedral meshing method is much smaller than the simulation result of the mixed meshing method. Figure 3 shows the center section of tetrahedral and hybrid grids, both of which are 20 mm in size and have a total of about 1.2 million grids. Figure 4 shows the outlet component monitoring diagram of the two meshes after a certain number of steps. It can be seen that the component fluctuation of tetrahedral meshing method is small, while the component fluctuation of hybrid meshes is still large. Finally, pure tetrahedral mesh is used in the simulation.

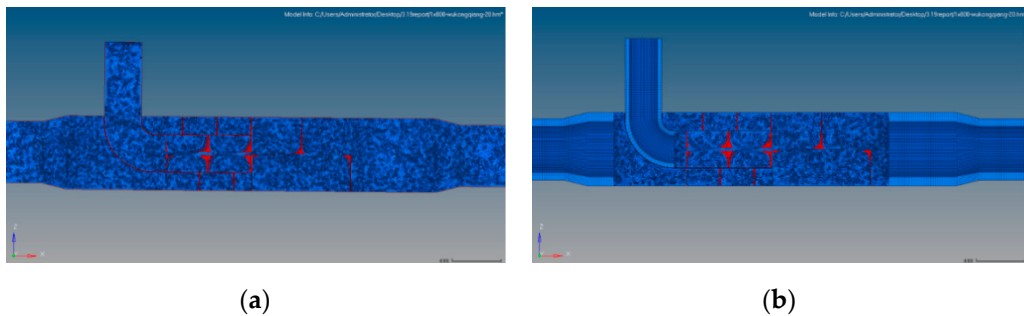

(**a**)  (**b**)

**Figure 3.** Center section of grid. (**a**) Tetrahedral network. (**b**) Hybrid networks.

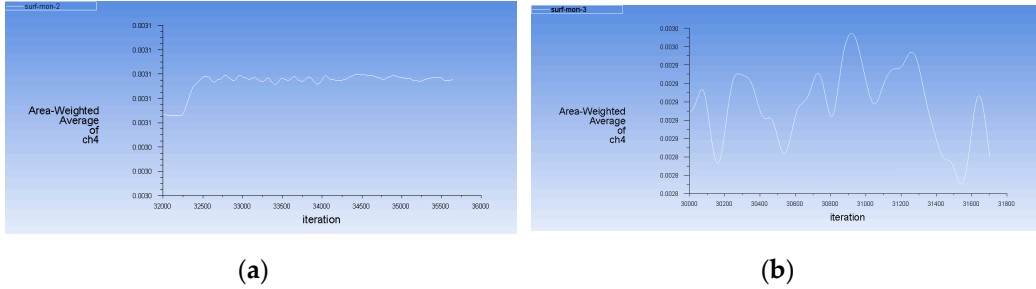

(**a**)  (**b**)

**Figure 4.** Monitoring curves of export components under the two meshing methods. (**a**) Tetrahedral network. (**b**) Hybrid networks.

After determining the partition method of tetrahedral mesh, we carry out the correlation check of mesh. The simulation results of different mesh sizes are calculated. By comparison, it is found that when the grid size is 15 mm, 20 mm, 25 mm and 30 mm, there is little difference in the concentration of the components at the outlet, indicating that the

increase in the total number of grids does not necessarily improve the accuracy of the calculation results. Finally, considering the simulation accuracy and calculation speed, the mesh size was determined to be 20 mm, and the total number of grids was approximately 1.2 million.

*2.3. Digital Analog Model*

In the process of solving with ANSYS fluent, the simple algorithm and separated implicit solver are used to solve the control equation, and the discretization is realized by the second-order upwind difference scheme [21]. The convergence criteria for all governing equations are set to be less than $10^{-6}$. In the iterative calculation process, whether the iteration is completed is determined by monitoring the fluctuation of the methane volume-fraction curve 1 m away from the outlet flange in the fluid area. When the monitoring result remains constant after 5000 steps of iterations, it is considered that the calculation and solution process can be considered converged. The governing equations used in the calculation process include continuity equation, momentum equation, energy equation and component transport equation, and their expressions are as follows:

Continuity equation:

$$\frac{\partial}{\partial x_i}(\rho u_i) = 0 \tag{1}$$

where $\rho$ is the density of the mixture and $u_i$ is the velocity component in the $i$ direction, $x_i$ is the coordinate component in each direction.

Momentum conservation equation:

$$\frac{\partial}{\partial x_j}(\rho u_i u_j - \tau_{ij}) = -\frac{\partial p}{\partial x_i} \tag{2}$$

where $p$ is pressure and $\tau_{ij}$ is shear stress.

Energy conservation equation:

$$\frac{\partial}{\partial x_j}(\rho u_j h + F_{h,j}) = u_j \frac{\partial P}{x_j} + \tau_{ij}\frac{\partial u_i}{\partial x_j} \tag{3}$$

where $h$ is the static total enthalpy and $F_{h,j}$ is the energy flux in the $x_j$ direction.

Component transport equation:

$$\frac{\partial}{\partial x_j}(\rho u_j m_l + J_{l,j}) = R_l \tag{4}$$

In the component transport equation, $m_l$ is the mass fraction of component $l$, $J_{l,j}$ is the diffusion flux of component $l$ in the direction, and $R_l$ is the formation rate of the chemical reaction of component $l$.

The resistance $\Delta P$ of the gas-mixing device is the pressure difference between the inlet 1 and the outlet section, including local resistance loss and resistance loss along the way. The local resistance loss is mainly caused by collision and vortex. In the calculation process, the resistance in the expanding area of the inlet section, the resistance in the shrinking area of the outlet section and the resistance of the spiral blade are primarily considered [22]. The uniformity $U$ of the gas-mixing device is calculated by the following formula:

$$U = 1 - \sqrt{\frac{1}{n}\sum_{i=1}^{n}(\frac{c_i - \overline{c}}{\overline{c}})^2} \tag{5}$$

where $n$ is the number of values of methane concentration on the exit section (uniformly distributed points), $c_i$ is the methane concentration at each point (molar fraction, the same below), and $\overline{c}$ is the average value of methane concentration on the section.

In engineering, when calculating the influence of coalbed-methane flow on the uniformity and resistance of the mixing device, the Reynolds number, Re, is often used to measure the change in fluid flow. The Reynolds number is a dimensionless number, and its physical meaning is the ratio of inertial force to viscous force in the flow field, which is calculated according to the following formula:

$$Re = \frac{\rho u d}{\mu} = \frac{4\rho Q_V}{\mu \pi d} \tag{6}$$

where $\rho$ and $Q_V$ are the density and volume flow of coalbed methane at 101,325 Pa and 0 °C respectively; $u$ is the average axial velocity of the section; $d$ is the section diameter and $\mu$ is the dynamic viscosity.

The boundary conditions of the calculation area mainly include the inlet, outlet and inner wall. The specific parameters are set as follows:

(1)    Inlet boundary condition

The inlet boundary conditions of the device are defined by the velocity inlet, and the mole fractions of methane and air at different inlets are given.

(2)    Exit boundary condition

It is considered that the flow has been fully developed on the outlet boundary of the calculation domain, and the outlet area is far from the reflux area. The pressure at the outlet is taken as the pressure value of the grid point on the upstream layer, and other physical quantities are taken as the value of the grid point on the upstream layer.

(3)    Solid wall condition

No slip condition is satisfied on the fixed wall, i.e., speed; the pressure is taken as the second type of boundary condition, i.e., $\partial p / \partial n = 0$.

In the construction of this numerical calculation model of the gas-mixing process, the selection of the turbulence model directly affects the convergence and accuracy of calculation results. By comparing the cloud images of methane component distribution at the outlet section calculated by the k-epsilon model, k-omega model and S-A model, it is found that when using the S-A model, the velocity curve has good convergence, and the internal flow field, velocity field and concentration field of the gas-mixing device show a more reasonable change process. Therefore, after comprehensive comparison, this paper adopts the S-A turbulence model for its calculations.

## 3. Calculation Analysis and Optimization Design

### 3.1. Basic Characteristics of Gas-Mixing Process

The air-mixing device is a kind of follow-up flow mixer, in which one air source is used as the active air source, and the other air source reacts to the change in the active air source according to the preset volume flow ratio. Generally, the air source with the large flow is set as the active air source, while the air source with the small flow is set as the follow-up air source. The change in the flow of the active air source can be adjusted in real time by controlling the opening of the regulating valve on the follow-up air source pipeline.

In a gas-mixing device with a flow rate of 7000 Nm$^3$/h, the flow rate and concentration of the small tube are 5000 Nm$^3$/h and 5%, respectively, and that of the large tube are 2000 Nm$^3$/h and 15%, respectively. The calculation results of the following two cases are analyzed: (1) Air-pipe device, no spiral set in the three areas; (2) Spiral air-mixing device, four left-hand spiral blades set in the small pipe area, with a pitch of 1400 mm and a total length of 700 mm; no spiral set in the large pipe area; there are three right-handed spiral blades in the mixing area, with a pitch of 1000 mm and a total length of 1000 mm.

Figure 5 is a comparison diagram of turbulent viscosity of a gas-mixing device under the two cases above. Compared with the empty pipe with almost no change in turbulent kinetic energy, the gas-mixing device mixes the coalbed methane of different concentrations in rapid flow through the installation of spiral blades in large pipe area, small pipe area and

mixing area. After strong collision and diffusion between different gases, turbulent flow is formed at the helical blade, and a mixed-gas flow with increasing energy is formed, so as to realize the rapid and uniform mixing of different gases. Before entering the mixing zone, the fluid is in a stable flow state. After entering the mixing zone, it is twisted and squeezed by the guide vanes. After centrifugation, the fluid velocity increases continuously. When the fluid gradually adapts to the deformed channel, the flow rate decreases continuously.

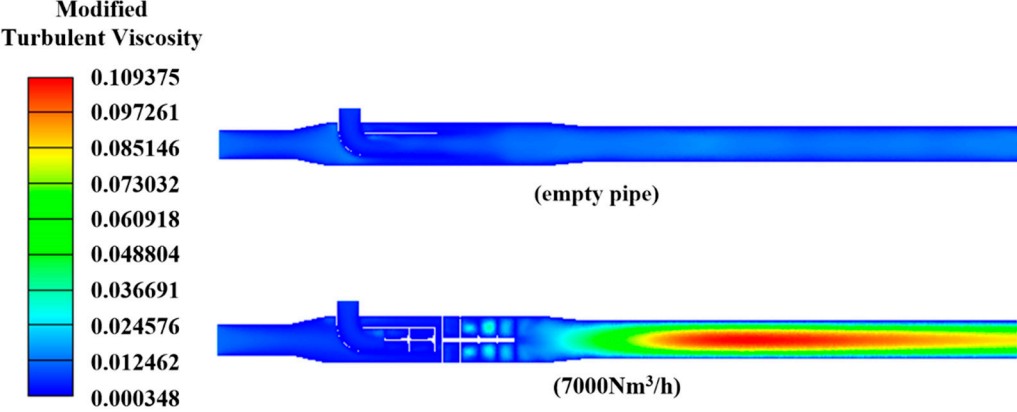

**Figure 5.** Comparison diagram of turbulent viscosity of the device.

Figure 6 is a comparison diagram of the internal pressure of the gas-mixing device under the above two cases. The pressure loss caused by the empty pipe is mainly loss along the way in the process of fluid flow. For the empty pipe device, the resistance loss can be almost ignored. In addition to the inevitable loss along the way in the process of fluid flow, there are other important pressure-loss factors in the spiral mixing device. Due to the existence of the guide vane of the device, the fluid collides with the vane in the process of fluid flow. The guide spiral blades of the large pipe area and the small pipe area rotate in the opposite direction, making the two gases mix more violently in the mixing area, resulting in the overall change in the pressure field of the device and the increase in the resistance loss before and after the gas-mixing device.

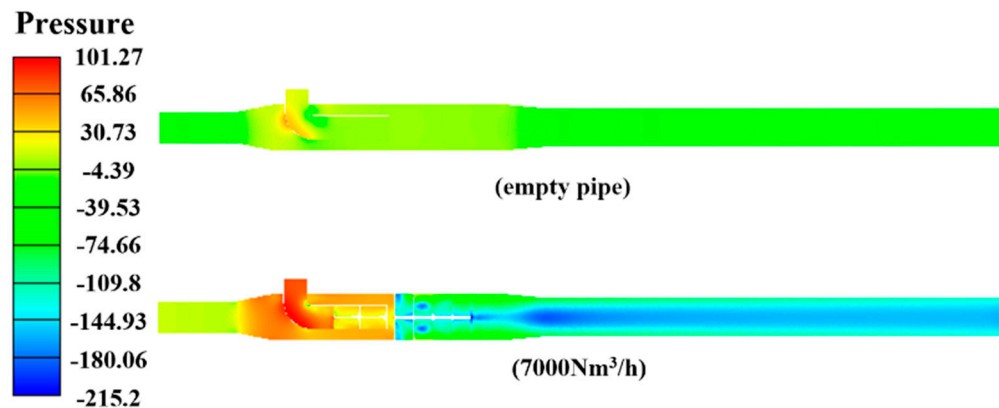

**Figure 6.** Comparison diagram of pressure loss.

Figure 7 shows the cloud diagram of methane component distribution in the central section of the device. Because the air pipe device is not equipped with a diversion device, the methane gas from the small pipe cannot be quickly and fully mixed with other gases. The component cloud map on the cross section 1 m away from the outlet shows that the methane concentration presents obvious stratification. For the spiral-mixing device, the reverse airflow in the large area and the small tube area will generate violent collisions between the molecules through the mixing zone. The centrifugal effect of the gas through the helical blades of the mixing zone renders the gas fully mixed two times. The existence of the helical blade can greatly shorten the distance of gas mixing.

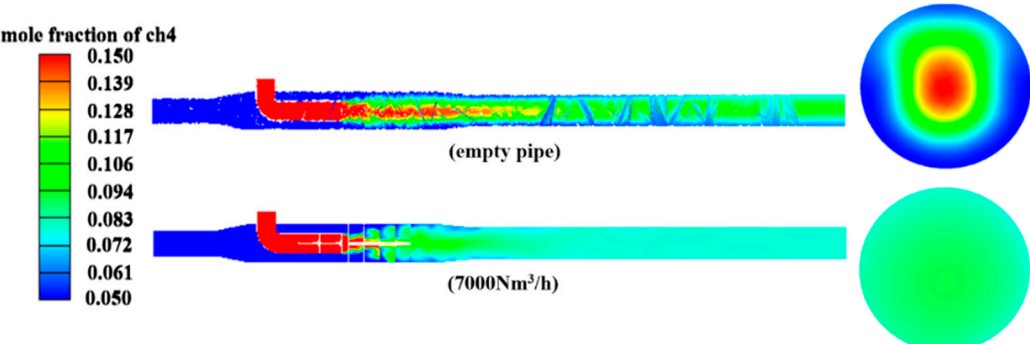

**Figure 7.** Cloud diagram of mixed gas components of empty tube and spiral tube.

*3.2. Spiral Distribution Analysis*

In the variable analysis of spiral structure, the first step is to determine the overall spiral distribution mode. This work mainly studies whether spiral blades are set in different areas and the influence of blade rotation direction on the uniformity of gas mixing and pressure loss during the flow of the device. In addition, because the change in device diameter and flow will affect the combined distribution of spiral blades in different areas, we set three sets of coalbed-methane-mixing devices with different flows and analyzed the spiral distribution mode.

When simulating the air mixing of the helical blade combination structure of the three sets of devices, it is necessary to keep other parameters of the device unchanged. Figure 8 shows the cloud diagram of methane-concentration distribution at 1 m from the outlet of the gas-mixing device with a gas flow of 7000 $Nm^3$/h under different spiral structures. The best combination of spiral distribution position and spiral rotation direction in the three areas of the device is analyzed from the results of the cloud diagram display. In the figure, R represents the right rotation of the spiral, L represents the left rotation of the spiral, and N represents that the spiral is not set. This corresponds to the small tube area, the large tube area and the mixing area from left to right. The specific comparative data are shown in Table 1.

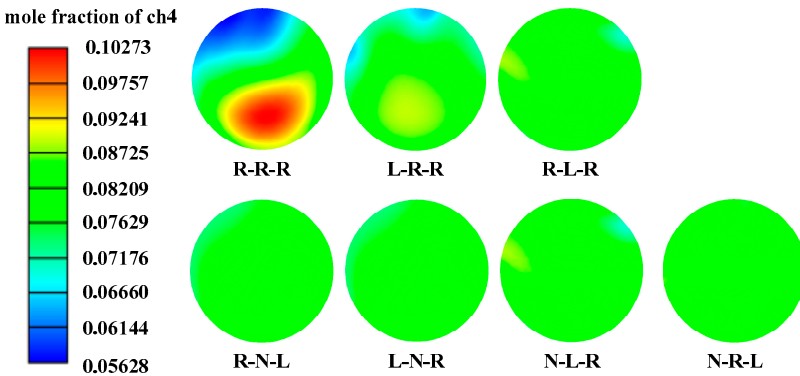

**Figure 8.** Methane concentration distribution at the outlet of gas-mixing devices with different spiral structures (7000 $Nm^3$/h).

From Figure 8 and Table 1, it can be seen that the combination of different spiral structures has a great impact on the resistance loss and gas-mixing uniformity of the unit, and the pressure loss of the unit is the largest when the outlet methane concentration distribution is the most uniform. Among them, when the L-N-R combination mode is adopted (left rotation in small pipe area, pitch 1400 mm, 0.5 turns and four pieces; no spiral in large pipe area; right rotation in mixing area, pitch 1000 mm, one turn and three pieces), the gas-mixing uniformity of the device is the highest, reaching 97.8%, and its pressure loss is only 81 Pa, which is the best choice in all kinds of combinations. For this arrangement, the direction of coalbed-methane turbulence in the small pipe area is opposite to that in the

mixing area, and the coalbed methane in the small pipe area rushes into the mixing area, and the flow direction changes greatly. In the inlet section of the mixing zone, the large pipe outlet occupies a larger section area outside, while the small pipe outlet is located inside and smaller than the area of the large pipe outlet. Therefore, the change in turbulence direction in the small pipe area is more favorable for increasing uniformity. In general, the flow rate of the large pipe area is much greater than that of the small pipe area, and its resistance plays a decisive role in the resistance of the air-mixing device. However, the large pipe area with this kind of arrangement has no blades, so the resistance is the least.

**Table 1.** Comparison of simulation results of performance parameters of gas-mixing devices with different spiral structures (7000 $Nm^3$/h).

| Combination Form | Resistance Loss $\Delta P$ (Pa) | Uniformity of Gas Mixing $U$ |
|---|---|---|
| R-R-R | 77 | 0.834864178 |
| L-R-R | 85 | 0.938966162 |
| R-L-R | 108 | 0.985306644 |
| R-N-L | 78 | 0.977691419 |
| L-N-R | 81 | 0.978119616 |
| N-L-R | 143 | 0.969055794 |
| N-R-L | 137 | 0.968450126 |

In the gas-mixing device with a flow rate of 50,000 $Nm^3$/h, the flow rate and concentration in the small pipe area are 45,000 $Nm^3$/h and 0% (air), respectively, and the flow rate and concentration in the large pipe area are 5000 $Nm^3$/h and 10%, respectively. The average concentration of methane after mixing is 1%. The mixing simulation results of the three-section spiral combination structure of the mixing device are shown in Table 2. Compared with the flow of 7000 $Nm^3$/h, the combination mode of different spiral structures has a greater impact on the gas-mixing uniformity of the device, in which the maximum gas-mixing uniformity is 92.64%, while the minimum gas-mixing effect can only reach 9.8%. The data in the table show that the uniformity of L-L-R and N-L-R combination gas mixtures is above 90%, and the pressure loss is close, while N-L-R is easier to process than L-L-R; thus, when the coalbed methane flow is 50,000 $Nm^3$/h, the N-L-R combination is finally selected, that is, there is no spiral in the small pipe area, left rotation in the large pipe area and right rotation in the mixing area, and the uniformity of gas mixing is 90.76% and the resistance loss is 286 Pa.

**Table 2.** Comparison of gas-mixing simulation results of three-stage spiral combined structure (50,000 $Nm^3$/h).

| Combination Form | Resistance Loss $\Delta P$ (Pa) | Uniformity of Gas Mixing $U$ |
|---|---|---|
| L-L-L | 108 | 0.098636925 |
| L-L-R | 296 | 0.926471529 |
| L-R-L | 277 | 0.879708554 |
| L-R-R | 115 | 0.144680942 |
| N-L-L | 110 | 0.12551543 |
| N-L-R | 286 | 0.907642821 |
| L-N-L | 145 | 0.742540376 |
| L-N-R | 156 | 0.797162208 |

In the gas-mixing device with a flow rate of 160,000 $Nm^3$/h, the flow rate and concentration of the small pipe are 150,400 $Nm^3$/h and 0% (air), respectively, and the flow rate and concentration of the large pipe are 9600 $Nm^3$/h and 10%, respectively. The average concentration of methane after mixing is 0.6%. Table 3 shows the simulation results of three-stage spiral structure of the gas-mixing device when the coalbed methane flow is 160,000 $Nm^3$/h. The device is applied to the feed-gas transmission pipeline of the heat-

storage oxidation device, and the average concentration of methane on the section is 1%. Among them, using the distribution mode of right rotation of blades in small pipe area, left rotation of blades in large pipe area and right rotation of blades in mixing area, the uniformity of gas mixing at the outlet can reach 98.24%, but the pressure drop increases to 1268 Pa. For other combinations with methane-mixing uniformity of more than 90%, the pressure loss is less than 500 Pa only when combined with L-N-R spiral structure. That is, this spiral distribution combination mode has a gas-mixing uniformity of more than 90% and minimum pressure loss. Therefore, comprehensively considering the pressure drop and gas-mixing uniformity of the unit, when the coalbed methane flow is 160,000 Nm$^3$/h, and the spiral arrangement of spiral left rotation in the small pipe area, no spiral in the large pipe area and right rotation (L-N-R) in the mixing area is adopted during the processing of the gas-mixing unit, the resistance loss is 498 Pa and the gas-mixing uniformity is 91.46%.

**Table 3.** Comparison of gas-mixing simulation results of three-stage spiral combined structure (160,000 Nm$^3$/h).

| Combination Form | The Resistance Loss $\Delta P$ (Pa) | Uniformity of Gas Mixing $U$ |
|:---:|:---:|:---:|
| R-R-R | 532 | 0.912529119 |
| R-L-R | 1268 | 0.98240801 |
| R-R-N | 520 | 0.961878988 |
| R-L-N | 525 | 0.9435166 |
| R-N-R | 471 | 0.69367898 |
| L-N-R | 498 | 0.914625259 |
| N-R-R | 534 | 0.827186138 |

*3.3. Influence of Gas-Mixing Device on Operating Flow*

After the air-mixing device is processed and installed according to the designed structure, its structure and size cannot be changed again, but due to the influence of the gas drainage system, the flow of coalbed methane may change. In order to ensure the stability and safety of the operation of the downstream coalbed-methane-utilization device, even if the flow changes, the uniformity of the gas concentration after coalbed-methane mixing must be ensured, and the resistance must meet the requirements.

The performance parameters of the mixing device used on the gas-source transmission pipeline of the coalbed-methane power-generation device are calculated. The rated flow of the mixing device is 7000 Nm$^3$/h. The influence of the change in the operating flow of the device on the uniformity of gas mixing and resistance loss is investigated. Figure 9 is the cloud diagram of the concentration distribution of methane at the outlet under different flow rates reflected by the Reynolds number, and Figure 10 shows the variation curve of gas-mixing uniformity and resistance loss of the device under the corresponding Reynolds number *Re*. It can be seen that the resistance loss of the device increases linearly with the increase in flow, and the uniformity of gas mixing basically changes little in the range of Reynolds number 275,000–375,000, but increases slightly after 375,000. From the numerical simulation results, the improvement of uniformity can be ignored, so the designed gas-mixing device can still meet the requirements of gas-mixing uniformity when the flow decreases. Because the structure and size of the device do not change, it is inevitable that the resistance loss increases with the increase in Reynolds number. With the increase in gas kinetic energy, more kinetic energy is lost due to the existence of resistance, that is, the resistance loss of the device increases with the increase in gas flow.

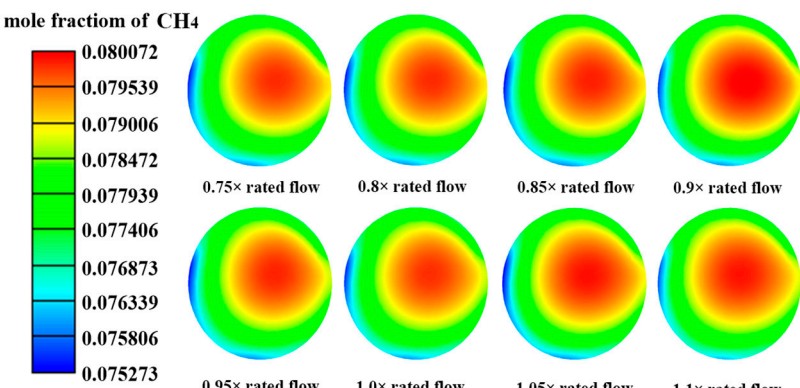

**Figure 9.** Effect of unit flow change on outlet methane composition.

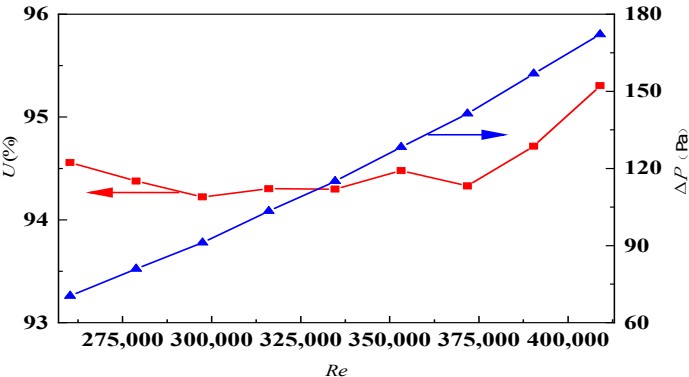

**Figure 10.** Influence curve of device flow change on device flow parameters.

## 4. Comparison of Experimental Results

In order to evaluate the accuracy of the above calculation results, experimental verification is needed. According to the numerical simulation results, three sets of coalbed-methane-mixing devices are processed with rated flow of 7000 Nm$^3$/h (for power generation of coalbed methane), 50,000 Nm$^3$/h (for coalbed-methane thermal storage oxidation) and 160,000 Nm$^3$/h (for heat storage and oxidation of coalbed methane). After processing, they are installed on the air inlet pipe at the front end of the coalbed-methane-utilization device. Differential pressure transmitters are arranged at the inlet and outlet of the mixing device to measure the resistance loss P of the device. In the test system, the uniformity is calculated from the methane concentration at different radial positions on the outlet section. As shown in Figure 11, the methane concentration sensor is arranged on the same section of the mixer outlet, and the sampling pipe is used to go deep into the mixer for sampling. The air intake of the sampling pipe is also evenly distributed on the centerline of a section. The number of methane concentration sensors can be flexibly set according to the diameter of the mixer. Limited by the cost, each mixing device is equipped with six sets of methane-concentration sensors in this experiment. After measuring the methane concentration at each point, the uniformity can be calculated according to Equation (5).

After the stable operation of the coalbed-methane-utilization unit, the resistance and uniformity of three units under rated flow were tested and compared with the calculated values (as shown in Table 4). It is found that there was a certain error between the calculated values and the measured values.

Through analysis, it was found that the roughness of the wall surface and the processing technology of the spiral blade should be the key factors causing resistance error in the experimental device. Increasing the wall roughness can improve the pressure drop of the device. Ordinary carbon steel was used in the processing of the air-mixing device, and the wall roughness was set at 0.06 in the calculation, which may be different from the actual value of the device, as shown in Figure 12. Therefore, in the process of manufacturing and

assembly, the change in blade-flow section angle and the reservation of necessary clearance in the process of processing and manufacturing will increase the measured pressure loss of the device, and these factors are very difficult to fully reflect in the numerical simulation.

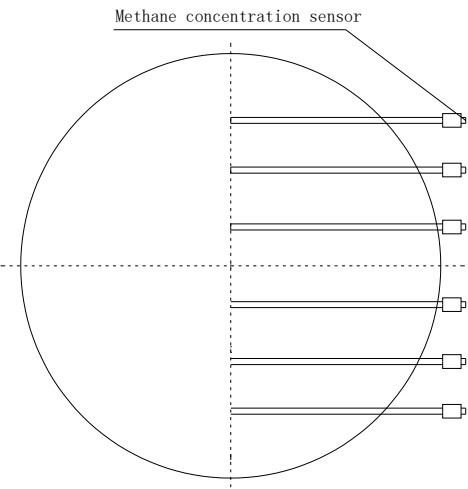

**Figure 11.** Layout of uniformity measuring points.

**Table 4.** Comparison between calculated and measured results of gas-mixing device.

| Flow (Nm³/h) | Combination Form | Calculated ΔP (Pa) | Measured ΔP (Pa) | Calculated U | Measured U |
|---|---|---|---|---|---|
| 7000 | L-N-R | 81 | 94 | 97.80% | 99.20% |
| 50,000 | N-L-R | 286 | 325 | 90.76% | 98.80% |
| 160,000 | L-N-R | 498 | 485 | 91.46% | 97.20% |

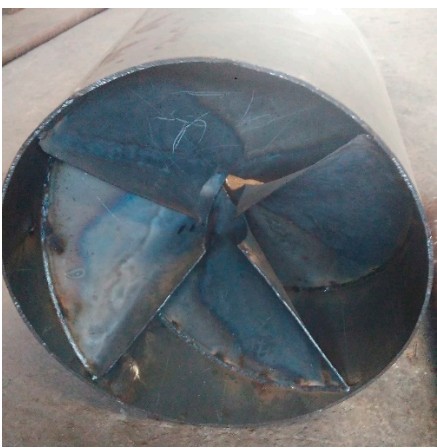

**Figure 12.** Arrangement of spiral blades in the experimental unit.

A coalbed-methane-utilization unit requires that the resistance of the mixing unit is not more than 500 Pa and the uniformity is not less than 90%. It can be seen from Table 1 to Table 3 that it is generally difficult to meet the above requirements if the screw structure is not properly arranged. In the calculation process, the arrangement of the above spiral structure is optimized appropriately. Although there are errors between the calculated results and the experimentally measured values, they all meet the requirements of resistance and uniformity of the coalbed-methane-utilization unit to the mixing unit. Therefore, numerical simulation can provide technical measures to reduce the resistance of the unit and improve the uniformity of coalbed-methane mixing with different concentrations, and

provide a stable gas source and safety guarantee for the downstream coalbed-methane-utilization unit.

## 5. Conclusions

(1)   A three-dimensional calculation model of the internal flow of the gas-mixing device is established, and the coalbed-methane-mixing process is numerically simulated. A tetrahedral mesh generation method is adopted in the gas-mixing device, with a size of 20 mm and a total mesh of about 1.2 million. The SIMPLIE algorithm is used to solve the governing equations, which is discretized by the second-order upwind difference scheme, and the S-A model is used for turbulence.

(2)   The guide vane is set inside the air mixing device, which can make the fluid produce a centrifugal effect after violent collision with the vane, shorten the mixing distance of gas and improve the uniformity of air mixing; However, the violent collision of two streams of coalbed methane in the mixing area will lead to the overall change in the internal pressure field of the device, thus increasing the resistance.

(3)   The screw distribution pattern of three sets of devices with different flow rates are determined, and the influence of different screw structure combination patterns on the flow uniformity and pressure loss of the device are studied. On this basis, the best spiral structure combination is optimized for three sets of devices with different flow rates. When the flow rate is 7000 $\text{Nm}^3/\text{h}$, 50,000 $\text{Nm}^3/\text{h}$ and 160,000 $\text{Nm}^3/\text{h}$, the best spiral structure combination modes are L-N-R, N-L-R and L-N-R, respectively.

(4)   The experimental results show that there are some errors between the numerical simulation and the experimental results. The main reason is that there are defects in the processing technology of the spiral blade structure of the experimental device. However, the purpose of this paper is to provide technical means to reduce the resistance of the gas-mixing device, improve the uniformity of coalbed-methane mixing with different concentrations, and provide stable gas sources and safety measures for coalbed-methane-utilization devices.

**Funding:** This research was funded by Chongqing Talent Plan innovation and entrepreneurship leading talent project (CQYC201903009) and China National Science and Technology Major Project (2016ZX05045-006). The APC was funded by Chongqing Talent Plan innovation and entrepreneurship leading talent project (CQYC201903009).

**Institutional Review Board Statement:** This article does not cover human or animal studies.

**Data Availability Statement:** The data that support the findings of this study are available from the corresponding author.

**Acknowledgments:** This work was supported by Chongqing Talent Plan innovation and entrepreneurship leading talent project and China National Science and Technology Major Project.

**Conflicts of Interest:** The author declared that he have no conflict of interest in this work.

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
