# Peer review of "Numerical Simulation and Experimental Analysis of Dynamic Continuous Operation of Low-Concentration Coalbed-Methane-Mixing Device"

_processes, doi:10.3390/pr10071265_

Round 1

Reviewer 1 Report

Paper entitled “Numerical simulation and experimental analysis of dynamic
continuous operation of low concentration coalbed methane mixing device” meets the necessary standards for publication in this journal.

Please check the entire manuscript carefully for eventual typographical errors.

Attention when writing references. They are not unitary.

Final Conclusion: The paper meets the necessary standards for publication.

Author Response

 Dear Editor and Reviewers,

The authors would like to thank Editor and Reviewers for comments to improve the quality of the paper. Your comments will be of great help to our future thesis writing. We have studied comment carefully and have made correction. The revised content has been marked in the text. You may need to use the "show comments" function under "review" in Microsoft Word. Looking forward to your approval.  

Thank you.

Paper entitled “Numerical simulation and experimental analysis of dynamic continuous operation of low concentration coalbed methane mixing device” meets the necessary standards for publication in this journal. Please check the entire manuscript carefully for eventual typographical errors.

Reply: The paper has been revised with the help of native English teachers. See the red part of the revised draft.

Reviewer 2 Report

The English translation must be improved. It needs a major revision.

The numerical simulation method must be better described:

a. The concentration at the outlet of the mixing device in Fig. 1 does not agree with the range given neither for power generation (10-30% outlet) nor for energy storage oxidation (1.2% outlet) -clarify

b. Clarify/explain: how is the concentration controlled in the device?  

c. What is the UG? Reference?

d. The "reflux" (p.4, line126-127) is a problem with the helix model approach. "The vortex generated by the gas mixing in the pipeline is not completely in the computational domain at the outlet" - what does this mean to the validity of the simulation? 

e. The device in Fig.1 is 3 m long, but the model is extended to 6 m long - what does this mean to the model results versus experimental results? Was it 6 m long?

f. The governing equations are a set of standard formulas from literature - why are they listed if these are internal to ANSYS? What had to be done to them? Selected? Modified? Or, just used as they are?

g. The boundary conditions are not defined in the internal zone from the exit of the spiral in the small pipe to the entrance to the large spiral region. Is there a spiral before the small pipe in the large pipe? A better figure is needed. 

h. The small tube acts as an injector booster for flow in the large pipe. What are the cross-sectional relations? A better figure for the dimensions are needed fro the interior area of the mixing zone.

i. The experimental results show acceptable agreement with numerical results for pressure loss but poor agreement (~10% difference) for mixing uniformity. In view of this, mixing uniformity should not be used that strictly (as to the 3rd or 4th decimal) when comparing arrangements for best arrangement selection. 

j. R-N-L or L-N-R arrangements should be given the same results if the Coriolis force field is ignored due to the rotation of the Earth. One table shows this, worth commenting.

k. This R-N-L is likely the best arrangement for all intake flow rates on common sense and physical principle considerations. Please comment why.

l. Three different main flow rates are defined for three cases but the flow rates in the small pipe and the concentrations are missing. Separate these three cases in separate points in the discussion of the cases and the results.

m. Fig. 7 is not clear - it needs a better presentation.

n. Fig. 10 is not clear - it needs a better picture.

Author Response

 Dear Editor and Reviewers,

The authors would like to thank Editor and Reviewers for comments to improve the quality of the paper. Your comments will be of great help to our future thesis writing. We have studied comment carefully and have made correction. The revised content has been marked in the text. You may need to use the "show comments" function under "review" in Microsoft Word. Looking forward to your approval.  

Thank you.

Reviewer 3 Report

The study is related to the end analysis of dynamic continuous operation of a low-concentration coalbed methane mixing device.

The topic is interesting, but the manuscript needs to be improved before consideration for later publication. The following issues have to be addressed:

- some sentences are too long which makes work less attractive to reader, e.g. first sentence in the abstract, please divide this,

- There are some repetitions of the same word, phrases in the same sentence, e.g., line 19 - "gas mixing" please rephrase it,

 after reading the text, I think that the text should be corrected by the language correction service,

- Please change the first sentence in the introduction, this is obvious,

- line 40 - In the coalbed methane utilization mode [5]. ??

- line 51,52 sentence Coal mining enterprises are the major energy and carbon suppliers of the country - please exclude carbon suppliers, it is obvious,

- lines 95-100, repetition of the same sentence,

- please make review of previous works connected with the topic in the introduction part,

- My suggestion is to connect FIg 1 and 2 in one picture, and e.g. indicate parts of it as a) b),

- lines 116-118 repetitions of the same sentence,

- line 126, you wrote "we found" please use impersonal form as it was found eg.; if there is a we, where are the rest of the authors?

- resolutions of the figures are really low and it is unacceptable in this form,

- correct some units, pa- Pa lines - 312,313 etc.,

- There should be some more references to previous studies connected with the described topic.

Author Response

(The authors gave the same response as above.)

Reviewer 4 Report

This study utilizes the ANSYS package to simulate the hydrodynamic behavior of a gas mixing device by a CFD technique. Although the author claimed that a dynamic model was developed, I think it is a static simulation study.

Based on my comments in the attached file, I think this manuscript needs a modification higher than a major level to be published as a research article. Therefore, I reject the current version of the manuscript.

Author Response

 Dear Editor and Reviewers,

The authors would like to thank Editor and Reviewers for comments to improve the quality of the paper. Your comments will be of great help to our future thesis writing. We have studied comment carefully and have made correction. The revised content has been marked in the text. You may need to use the "show comments" function under "review" in Microsoft Word. Looking forward to your approval.  

Thank you. 

This study utilizes the ANSYS package to simulate the hydrodynamic behavior of a gas mixing device by a CFD technique. Although the author claimed that a dynamic model was developed, I think it is a static simulation study.

Reply: Coalbed methane mixing system is a set of intelligent mixing system including automatic control program. As shown in Figure 1, the concentration and flow rate of coalbed methane at the two entrances change frequently. In actual operation, the gas mixing system needs to adapt to the dynamic change of inlet gas concentration and change the flow rate through PLC program to ensure the stable concentration of coalbed methane at the outlet. See Q2 of the revised draft.

The variation of the concentration of imported coalbed methane is irregular, which is related to many devices, technology and management factors in underground coal mine, so the unsteady flow cannot be simulated.

Based on my comments in the attached file, I think this manuscript needs a modification higher than a major level to be published as a research article.

Reply: The paper has been greatly modified. Please refer to "peer-review-20051423.v1.pdf" for the annotations and revisions. The paper has been revised by the native English teacher. Please see the red part in the revised draft.

Round 2

Reviewer 2 Report

I accept the amendments and author's replies to my review comments.

Author Response

Ok, thank you for accepting my revised manuscript.

Reviewer 4 Report

Dear authors

I am a non-Chinese researcher and cannot read/understand Chinese texts.

Your added texts to the "peer-review-20051423.v1.pdf" are in Chinese, and I cannot check your revisions.

Furthermore, it is not a good idea to use the Ms-Word track change to show your revisions (some useless notations occupy almost half of the pages). Please simply highlight the changes made in the manuscript.

Please resolve this issue so that I can check your manuscript once again.

Author Response

 Dear reviewers,

I'm really sorry for the trouble. Now I will upload it again and look forward to your reply. I apologize for the inconvenience.

The paper has been revised by the native English teacher. Please see the red part .

As for your question, I have highlighted it in green in the "PDF-revised draft.pdf" file I uploaded.

Round 3

Reviewer 4 Report

Dear authors,

Thank you so much for completely addressing my concerns during the revision phase.

I confess the current version of your article is acceptable for publication in the Processes journal.

My congratulation.